# Impact of Magnetite Nanowires on In Vitro Hippocampal Neural Networks

**DOI:** 10.3390/biom13050783

**Published:** 2023-04-30

**Authors:** Belén Cortés-Llanos, Rossana Rauti, Ángel Ayuso-Sacido, Lucas Pérez, Laura Ballerini

**Affiliations:** 1Departamento de Física de Materiales, Universidad Complutense de Madrid, 28040 Madrid, Spain; 2Fundación IMDEA Nanociencia, C/Faraday 9, 28049 Madrid, Spain; 3Department of Medicine, Duke University, Durham, NC 27705, USA; 4International School for Advanced Studies (ISAS-SISSA), 34136 Trieste, Italy; 5Deparment of Biomolecular Sciences, Università degli Studi di Urbino Carlo Bo, 61029 Urbino, Italy; rossana.rauti@uniurb.it; 6Brain Tumor Laboratory, Fundación Vithas, Grupo Hospitales Vithas, 28043 Madrid, Spain; ayusoa@vithas.es; 7Faculty of Experimental Science and Faculty of Medicine, University of Francisco de Vitoria, 28223 Madrid, Spain

**Keywords:** nanowires, iron oxide, hippocampal neuronal networks, neuronal activity, live imaging

## Abstract

Nanomaterials design, synthesis, and characterization are ever-expanding approaches toward developing biodevices or neural interfaces to treat neurological diseases. The ability of nanomaterials features to tune neuronal networks’ morphology or functionality is still under study. In this work, we unveil how interfacing mammalian brain cultured neurons and iron oxide nanowires’ (NWs) orientation affect neuronal and glial densities and network activity. Iron oxide NWs were synthesized by electrodeposition, fixing the diameter to 100 nm and the length to 1 µm. Scanning electron microscopy, Raman, and contact angle measurements were performed to characterize the NWs’ morphology, chemical composition, and hydrophilicity. Hippocampal cultures were seeded on NWs devices, and after 14 days, the cell morphology was studied by immunocytochemistry and confocal microscopy. Live calcium imaging was performed to study neuronal activity. Using random nanowires (R-NWs), higher neuronal and glial cell densities were obtained compared with the control and vertical nanowires (V-NWs), while using V-NWs, more stellate glial cells were found. R-NWs produced a reduction in neuronal activity, while V-NWs increased the neuronal network activity, possibly due to a higher neuronal maturity and a lower number of GABAergic neurons, respectively. These results highlight the potential of NWs manipulations to design ad hoc regenerative interfaces.

## 1. Introduction

The application of nanotechnology to nervous system repair strategies has increasingly targeted the development of nanomaterials to interface neurons and neuronal networks [1,2,3]. However, the clinical application of nanotechnology-based recording and stimulation devices is still limited due to their material size penetration and how their interaction impacts neuronal network functionality [4,5,6]. Importantly, diverse nanomaterial dimensionality can elicit distinct responses in brain cells and circuits [7,8]. In this framework, particular attention must be given to nanowires (NWs) as a permissive substrate for neural growth [9,10]. NWs’ nano-size enables their crossing of the plasma membrane without the emergence of significant cell damage [11]. NWs have been used for positioning neurons [12], directing neurite growth [10,13,14], and extracellular and intracellular recording through neural cell membranes [15,16,17,18]. 

In recent years, NWs have been exploited in arrays by tuning their size and orientation to improve cell adhesion. Recent advances in nanotechnology have allowed further manufacturing of nanostructures with high aspect ratios, such as NWs or polymer patterns [19]. Using different fabrication technologies, novel standard integrated circuits based on NWs, namely scaffolds or multielectrode arrays (MEA), were fabricated to record neuronal network activity [17,20]. Among these techniques, electrodeposition is an electrochemistry approach that provides the precise control of NWs diameter and length via synthesis parameters [21]. The ease of tuning NWs orientation might be relevant in the design of devices for sensing and manipulating cells [14,22,23].

Together with NWs orientation, the building material may also affect functional cellular properties, such as adhesion and survival [24,25,26]. Platforms with nanomaterials presenting optical, magnetic, and mechanoresponsive properties have shown greater effects on neuronal recording and modulation [4,27]. High-density silicon (Si) and gallium phosphide (GaP) semiconductor NWs were shown to affect the morphology of retinal or peripheral neurons [24,28,29]. In recent decades, due to their electrical and mechanical properties, carbon-based materials have been successfully used as substrates for neuronal growth due to their ability to interact with neuronal membranes and synapses [30,31,32,33,34]. Nonetheless, depending on the material size, semiconductors and carbon-based materials might need a coating to increase their biocompatibility [35]. In this framework, iron oxide is a well-known biocompatible material [21,36,37]. Iron oxide nanoparticles (IONPs) are being used in many experimental settings in the field of regenerative medicine, including nervous systems diseases, and for cancer treatments, such as hyperthermia and magnetic resonance imaging (MRI) [37,38,39,40,41]. Among iron oxide materials, magnetite (Fe_3_O_4_) and maghemite (γ-Fe_2_O_3_) are particularly both biocompatible and biodegradable [21,37,41]. 

Although several studies describe the interaction between NWs and living cells and show the impacts on cell viability and network changes [10,24,25,42,43], very few investigate the ability of NWs orientation to impact network morphology and activity [26,29,44,45], i.e., gold NWs in a vertical orientation promoted neuronal maturation and network connectivity compared with a flat gold platform [44]. While taking advantage of the elongated aspect of NWs and their iron oxide properties, we studied the ability of iron oxide NWs orientation to interact with an in vitro hippocampal network for the first time. Electrodeposited iron NWs were thermally oxidized to obtain magnetite NWs. Scanning electron microscopy (SEM) was used for their morphological characterization, and a tensiometer was used for the wettability measurements. Raman spectroscopy was performed to characterize the iron oxide present in the samples. Hippocampal cultured neurons were interfaced with the NWs, and after 8–10 days in vitro (DIV), we investigated cell density using immunofluorescent labelling and confocal microscopy. Live calcium imaging was used to assess the functional properties of neurons and networks. NWs orientation can affect neuronal density and glial morphology, but more interestingly, it can impact emerging synaptic network activity.

## 2. Materials and Methods

### 2.1. NWs Synthesis

Iron NWs were electrodeposited inside the nanopores of a polycarbonate membrane supplied by Sterlitech. Before the electrodeposition, an Au thin film was thermally evaporated on one side of the template to use it as a working electrode. The electrolyte was composed of 0.5 M iron sulfate heptahydrate (Fe_2_SO_4_∙7H_2_O) and 0.5 M boric acid (H_3_BO_3_), both supplied by Panreac. The pH of the electrolyte was adjusted to a pH of 2.5 by dropping sulfuric acid (H_2_SO_4_) to the solution. The synthesis was carried out in a three-electrode vertical cell with a Pt mesh as a counter electrode and an Ag/AgCl electrode as reference. The Fe NWs were synthesized with a constant potential of −1.15 V (vs. Ag/AgCl) which is a growth rate of 12.5 nm/s. After electrodeposition, the template was removed using dichloromethane (CH_2_Cl_2_) from Sigma-Aldrich, and the NWs were deposited onto Si wafers or glass coverslips for characterization.

### 2.2. Physical Characterization

The morphology of the samples was studied using scanning electron microscopy (SEM) and a JEOL JEM 6335 microscope (Tokio, Japan). The static contact angle measurements were performed using an optical tensiometer (Attension Theta, Biolin Scientific, Manchester, UK). Deionized water drops of 5 μL were deposited onto the substrate. Different angles with the baseline (left and right) and mean values were calculated by adjusting the drop profile to a Young–Laplace curve. 

Raman measurements were carried out using a confocal Raman microscope (Witec ALPHA 300RA, OXFORD Instruments, Abingdon, UK) at room temperature with an Nd:YAG laser (532 nm). The spectra were measured at 0.25 mW of laser excitation power and used an objective with a numerical aperture (NA) of 0.95. Raman spectra were recorded in the spectral range of 0−3600 cm^−1^. We measured regions on the plane (XY scans, 5 × 5 µm^2^) and in-depth (XZ scans, 5 × 2 µm^2^). Punctual spectra were recorded along with the XZ, in-depth scans to identify the plane with larger Raman intensity. The spectra were analyzed using Witec Control Plus Software (version 2.08). The band positions obtained fitting as Lorentzian functions. 

### 2.3. Sterilization Protocol

Iron NWs were deposited in glass coverslips (Kindler, EU), and the template was removed using dichloromethane (CH_2_Cl_2_; Sigma-Aldrich, St. Louis, MO, USA). The samples were sterilized in an oven at 180 °C for 4 h to obtain sterilized magnetite NWs. Then, 0.1 mg/mL of poly-L-lysine-coated (PLL) (Sigma-Aldrich) was dropped into the samples and left for 7 h in an incubator at 37 °C (5% CO_2_). After that, we removed the PLL, and the coverslips were washed three times with sterilized water. The samples were left in the incubator overnight (37 °C; 5% CO_2_). 

### 2.4. Preparation of Primary Hippocampal Cultures

As previously reported, primary hippocampal cultures were prepared from postnatal day 2 or 3 (P2-P3) rats [31,46]. All procedures were approved by local veterinary authorities and performed following Italian law (decree 26/14) and UE guidelines (2007/526/CE and 2010/63/UE). Animal use was approved by the Italian Ministry of Health. All efforts were made to minimize suffering and to reduce the number of animals used. All chemicals were purchased by Sigma-Aldrich unless stated otherwise. Briefly, enzymatically dissociated hippocampal neurons were plated on three different substrates, including poly-L-lysine-coated, R-NWs, and V-NWs substrates, at a density of 150,000 cells/mL (n = 3 culture series). 

Cultures were incubated (37 °C; 5% CO_2_) in a medium consisting of MEM (Invitrogen, Waltham, MA, USA), supplemented with 35 mM glucose, 15 mM HEPES, 1 mM apo-transferrin, 48 µM insulin, 3 µM biotin, 1 mM vitamin B12, and 500 nM gentamicin (Gibco). The numbers of neurons and astrocytes were counted 1 day after plating to compare the relative amount of each cell type. In our growth condition, the neuron-to-astrocyte ratio was 1:1, with 48.54% ± 18.3% of neurons and 53.12% ± 20.2% of astrocytes (n = 3 culture series). The culture medium was renewed 2 days after seeding and thereafter changed every 2 days. Proliferation of non-neural cells was prevented by the addition of 10 μM arabinofuranosyl cytidine (Ara-C) from the second day in culture onward. Cultures were then used for experiments after 8–10 DIV.

### 2.5. Immunocytochemistry and Image Processing

After 8–10 DIV, hippocampal cultures were washed three times with PBS and post-fixed in 4% paraformaldehyde (PFA, prepared from fresh paraformaldehyde) in PBS for 20 min at room temperature (RT). After that, cells were washed three times with PBS and permeabilized with 1% Triton X-100 for 30 min, blocked with 5% FBS in PBS for 30 min at room temperature, and incubated with primary antibodies for 45 min. The primary antibodies used were rabbit polyclonal anti-β-tub III (1:500 dilution) and mouse monoclonal anti-GFAP (1:500 dilution). After PBS washes, cells were incubated for 45 min with AlexaFluor 594 goat anti-rabbit (Invitrogen, dilution 1:500) and AlexaFluor 488 goat anti-mouse (Invitrogen, Waltham, MA, USA, dilution 1:500). Samples were mounted in Vectashield with DAPI to stain the nuclei (Vector Laboratories) on 1 mm thick coverslips. Cells were imaged at 20 × (0.5 NA) magnification using a Leica DM6000 fluorescent microscope (Leica Microsystems GmbH, Wetzlar, Germany). Cell densities, β-tub III areas, and GFAP perimeters and shapes were quantified using a random sampling of seven to ten fields (713 × 533 µm^2^; control and NWs substrates, at least n = 3 culture series). 

For GABA staining, cells were fixed with 1% glutaraldehyde and 4% PFA in PBS for 1 h in darkness at RT. After three washes with PBS, cells were blocked with 5% BSA, 3% triton, and 1% FBS in PBS at RT and incubated with primary antibodies for 45 min. The primary antibodies were rabbit polyclonal anti-GABA (1:300 dilution) and mouse polyclonal anti-β-tub III (1:500 dilution). After PBS washes, samples were incubated for 45 min at RT with secondary antibodies, AlexaFluor 488 goat anti-rabbit (1:500 dilution), AlexaFluor 594 goat anti-mouse (1:500 dilution), and DAPI for the nuclei (1:500 dilution). Samples were mounted in Vectashield on 1 mm thick coverslips. Images were acquired using a Nikon C2 confocal microscope (Nikon, Tokio, Japan) at 40× magnification. Z-stacks were acquired every 500 nm from seven to ten random fields for the control and NWs substrates. Offline analyses of cell density, stellate cells, and GABAergic neurons were performed using the image-processing package Fiji. Stellate cells were defined when they possessed processes longer than their perinuclear diameters [47,48]. Offline analyses of β-tub III areas and GFAP perimeters were performed using Volocity software, Volocity 3D image and quantification analysis software, PerkinElmer, Waltham, MA, USA). The images were acquired using identical exposure settings for each set of experiments. For the β-tub III area and GFAP perimeter quantifications, the intensity threshold was defined and adjusted to 0.5 SD. The area or the perimeter within each ROI with an intensity above the threshold was calculated and used for statistics. 

### 2.6. Calcium Imaging

Cultures were loaded with cell-permeable Ca^2+^ dye Oregon Green 488 BAPTA-1 AM (Invitrogen). A stock solution (4 mM) of the Ca^2+^ dye was prepared in DMSO, and cultures were incubated with a final concentration of 4 µM for 30 min (37 °C; 5% CO_2_). The samples were then placed in a recording chamber mounted on an inverted microscope (Nikon Eclipse Ti-U) where they were continuously superfused at RT by a recording solution of the following composition (mM): 150 NaCl, 4 KCl, 1 MgCl_2_, 2 CaCl_2_, 10 HEPES, and 10 glucose (pH adjusted to 7.4 with NaOH). Cultures were observed with a 20× objective (0.45 NA), and recordings were performed from visual fields (512 × 512 µm^2^, binning 4). Ca^2+^-dye was excited at 488 nm with a mercury lamp, and the excitation light was separated from the light emitted from the sample using a 395 nm dichroic mirror and an ND filter (1/32). Images were continuously acquired (exposure time 150 ms) using an ORCA-Flash4.0 V2 sCMOS camera (Hamamatsu, Shizuoka, Japan). The imaging system was controlled by integrating the imaging software (HCImage Live). After 10 min of recording, 10 µM bicuculline methiodide was bath-applied. At the end of each experiment, tetrodotoxin (TTX, 1 µM, a voltage-gated, fast Na^+^ channel blocker; Latoxan, Portes-lès-Valence, France) was applied to confirm the neuronal nature of the recorded signals. Recorded images were analyzed offline with Fiji (selecting region of interest, ROI, around cell bodies) and Clampfit software (pClamp suite, 10.2 version; Molecular Devices LLC, San Jose, CA, USA). Intracellular Ca^2+^ transients were expressed as a fractional amplitude increase (ΔF/F_0_, where F_0_ is the baseline fluorescence level and ΔF is the rise over baseline). We determined the onset time of neuronal activation by detecting those events in the fluorescence signal that exceeded the standard deviation of the noise by at least five times.

### 2.7. Synchronization Analysis

The mean correlation coefficient of calcium transients was computed by calculating the cross-correlation matrix between all pairs of neurons. To do so, we used the xcorr function in MATLAB. The xcorr function measures the similarity between the calcium activity of a neuron (i) and the lagged trace of another neuron (j) as a function of the lag. The correlation coefficient is the cross-correlation value at lag = 0 at which the traces overlap perfectly. The traces of each sample were considered independently from other samples, and all traces were simultaneously analyzed.

### 2.8. Data Analysis and Statistics

All data are presented as mean ± SD of the mean (n is the number of cells unless otherwise indicated). Statistical significance was calculated as a function of control/R-NWs/V-NWs, and GraphPad software (version 9) was used to evaluate the differences among the group with one-way ANOVA tests. A value of *p* < 0.05 was accepted as indicative of a statistically significant difference (*p* < 0.05, *; *p* < 0.01, **; and *p* < 0.001, ***). In box plots, the thick horizontal bar indicates the median value, while the boxed area extends from the 25th to 75th percentiles, with the whiskers ranging from the 5th to the 95th percentiles.

## 3. Results

### 3.1. Synthesis and Characterization of NWs

Two different configurations were tested to address the issue of different NWs orientations on a functional brain network. Figure 1a,b shows SEM images of electrodeposited iron NWs in random (R-NWs) and vertical (V-NWs) positions on the substrate. The NWs had an average diameter of 150 nm and approximate lengths of 1μm. After the thermal treatment produced during sterilization, the NWs were oxidized to iron oxide. Figure 1c,d shows an optical image of R-NWs and V-NWs after sterilization, where we observed the homogeneous distribution of the NWs (black) along the substrate (yellow). Wettability measurements give us the degree of wetting when a solid (NWs substrates) and a liquid interact. Contact angle values for R-NWs and V-NWs were 103° and 135°, respectively. Iron oxide R-NWs presented higher wettability compared with V-NWs. 

### 3.2. Investigating the Hippocampal Network

We compared primary hippocampal neurons upon 8–10 DIV of growth on poly-L-lysine glass coverslips (named control) with those developed on R-NWs or V-NWs substrates. Immunofluorescence techniques were used to determine the cellular composition of the different substrates by imaging the specific cytoskeletal component β-tubulin III (β-tub III) to visualize neurons (Figure 2a–c) and glial fibrillary acidic protein (GFAP) to visualize astrocytes (Figure 2f–h) [31,46].

We compared neuronal density (estimated by quantifying β-tub III cells) among the three different growth conditions. We measured 182.4 ± 73.9 neurons/mm^2^ in the control (n = 59 visual fields, 8 samples, n = 3 series of cultures), R-NW 241.2 ± 74.4 neurons/mm^2^ (n = 62 visual fields, 7 samples, n = 3 series of cultures), and V-NWs 202.6 ± 115.0 neurons/mm^2^ (n = 59 visual fields, 8 samples, n = 3 series of cultures) (Figure 2d), with a significant increase in cell density in R-NWs substrates (*** *p* < 0.001 and * *p* < 0.05, Kruskal–Wallis one-way ANOVA) when compared with control and V-NWs cultures. Consistently, we observed a significant increase (** *p* < 0.01, Kruskal–Wallis one-way ANOVA) in the β-tub III area for R-NWs ((1.75 ± 0.24) × 10^5^ µm^2^, n = 69 visual fields, 7 samples) when compared with the control ((1.58 ± 0.36) × 10^5^ µm^2^, n = 65 visual fields, 8 samples). V-NWs displayed lower values of their β-tub III area ((1.58 ± 0.37) × 10^5^ µm^2^, n = 68 visual fields, 8 samples) when compared with R-NWs, although without reaching statistical significance (Figure 2e). 

Using a similar approach, we investigated the effect of the R-NWs and V-NWs substrates on glial cell density. We identified and quantified GFAP^+^ cells in the control (87.4 ± 26.8 GFAP cells/mm^2^, n = 45 visual field, 6 samples), R-NWs (119.1 ± 40.4 GFAP-cells/mm^2^, n = 46 visual field, 5 samples), and V-NWs (73.7 ± 34.8 GFAP-cells/mm^2^, 46 visual field, 6 samples). We observed a significant increase in the GFAP^+^ cells in the R-NWs substrates when compared with the control (** *p* < 0.01, Kruskal–Wallis one-way ANOVA) and with V-NWs (*** *p* < 0.001, Kruskal–Wallis one-way ANOVA) (Figure 2i). We further quantified the presence of glial cells displaying a stellate morphology (see methods) as a percentage of the stellate glial cells on the total number of GFAP^+^ cells. We detected (36.7 ± 25.9)% of stellate GFAP^+^ cells in the control (n = 43 visual fields, 6 samples), which was similar to the R-NWs (38.2 ± 32.3%, n = 44 visual fields, 5 samples), while the stellate morphology ratio was significantly increased by the V-NWs substrates (61.7 ± 26.0%, n = 43 visual fields, 6 samples; *** *p* < 0.001, Kruskal–Wallis one-way ANOVA) (Figure 2j). To further assess the GFAP^+^ cells’ features, we calculated the GFAP perimeter (Figure 2k). The values detected in the control (161.3 ± 48.1 µm, n = 51 visual fields, 6 samples), R-NWs (117.6 ± 59.6 µm, n = 50, 5 samples), and V-NWs (237.4 ± 165.8 µm, n = 44, 6 samples) conditions were in accordance with the ratio of stellate glial cells detected with high values in the V-NWs (* *p* < 0.05, Kruskal–Wallis one-way ANOVA vs. the control, *** *p* < 0.001, Kruskal–Wallis one-way ANOVA vs. the R-NWs). 

### 3.3. Exploring Neuronal Activity

To investigate the network dynamics of neuronal cells grown on R- and V-NWs, we monitored neuronal emerging activity using fluorescence calcium imaging (see methods) of representative fields (318.2 × 318.2 µm^2^). At 8–10 DIV, neurons were synaptically connected and displayed spontaneous activity, including irregular bursts of synchronized firing epochs [31,32,46]. Figure 3a–c snapshots show the recorded fields and spatial distribution of recorded cells simultaneously traced within the same field of view with single-cell resolution. We compared and characterized spontaneous Ca^2+^ oscillations (Figure 3d). Such activity was always fully blocked by tetrodotoxin (TTX, 1 μM) application [31,49]. 

In our recordings, spontaneous Ca^2+^ activity was detected in 70% of the cells grown on glass coverslips, 51% on R-NWs, and 79% on V-RNWs, detecting no significant differences between the control and magnetite substrates (Fisher’s exact test, n = 8 visual fields for control, n = 6 R-NWs, n = 6 V-NWs, and n = 3 series of cultures) (Figure 3e). We measured the occurrence of spontaneous Ca^2+^ episodes in active cells by quantifying the frequency distribution that was significantly (* *p* < 0.05, Kruskal–Wallis one-way ANOVA) reduced in the R-NWs (0.034 ± 0.051 Hz, n = 95 cells, 5 different samples) when compared with the control ones (0.051 ± 0.088 Hz, n = 273 cells, 8 different samples; plot in Figure 3f). Conversely, in V-NWs, the frequency of calcium events was significantly higher (0.096 ± 0.058 Hz, n = 506 cells, 5 different samples; *** *p* < 0.001, Kruskal–Wallis one-way ANOVA) when compared with both the control and R-NWs (Figure 3f), despite the lower neuronal density compared with the R-NWs. 

In the second set of experiments, we pharmacologically blocked GABA_A_ receptors by bicuculline (10 μM; 20 min) application. In neural networks, the removal of the GABAergic synaptic component is known to alter the emerging activity patterns [50,51,52], leading to more intense and regular bursting [50,53,54]. In Figure 3g, fluorescent tracings show the appearance of Ca^2+^ episodes brought about by bicuculline in active cells. Upon this treatment, the percentage of active cells increased in all conditions without significant differences between the three substrates: (98 ± 2)% in the control (n = 7), (98 ± 2)% in the R-NWs cultures (n = 4), and (57 ± 38)% (n = 5) in the V-NWs ones (Figure 3h). Moreover, the frequency of calcium events increased to 0.148 ± 0.058 Hz (n = 430 cells from 7 different samples) in the control cultures. This result was not significantly different when compared with the R-NWs cultures (0.148 ± 0.097 Hz; n = 119 cells from 4 different samples) but was significantly (*** *p* < 0.001, Kruskal–Wallis one-way ANOVA) higher when compared with the V-NWs (0.110 ± 0.030 Hz, n = 497 cells from 4 different samples). Neurons were visually identified and functionally confirmed as neurons by the application of tetrodotoxin (TTX) (a voltage-gated, fast Na^+^ channel blocker) at the end of each measurement to confirm the neuronal nature of the recorded signals (Appendix A). The frequency of calcium events decreased in all substrates to 0.006 ± 0.005 Hz (n = 172 cells, 7 samples) in the control cultures which was not significantly different when compared with the R-NWs cultures (0.005 ± 0.004 Hz, n = 118 cells, 4 samples) and V-NWs cultures (0.006 ± 0.006 Hz, n = 73, 4 samples).

In summary, regardless of the diverse neuronal cell density, we did not detect differences in the number of active cells among the three substrates. Nevertheless, cells in the V-NWs were significantly more active when compared with the control and R-NWs in a standard medium. Still, the difference was reverted upon pharmacological removal of GABA_A_ receptor-mediated activity. This prompted us to investigate and compare the number of GABAergic neurons among all conditions to inspect the network contribution of this cell phenotype. Neurons were co-immunostained with antibodies for β-tub III and anti-GABA (Figure 4). We identified GABAergic cells as double immunolabeled neurons. We detected a significant (*** *p* < 0.001, Kruskal–Wallis one-way ANOVA) reduction in their number in the V-NWs cultures ((9 ± 5)%, n = 29 visual fields from 3 different samples) compared with the control ((15 ± 9)%, n = 30 visual fields from 3 different samples; see the plot in Figure 4d). No significant differences were found when comparing control or V-NWs cultures with R-NWs ones ((12 ± 9)%, n = 28 visual fields from 3 different samples). 

## 4. Discussion

In this study, we reported the ability of NWs orientation to impact neuronal growth and functionality. Nanowires’ biocompatibility depends on their diameter, length (a few microns), and the building material [24,25,55]. Furthermore, studies proved Fe_3_O_4_ biocompatibility [36]. Fe_3_O_4_ NWs platforms with lengths of 1 μm did not present any viability issue. The differences regarding the contact angle values between R-NWs and V-NWs were due to their orientation. Orientated NWs substrates (V-NWs) possess contact angle values closer to superhydrophobic materials as their contact angle values are about 150° due to their higher aspect ratio [56]. These changes in the aspect ratio could affect cell adhesion and its network. 

We observed that NWs orientation plays an important role in modulating the morphology of the neural cellular network. Accordingly, when we used V-NWs, the morphology of the neural cell network was not affected, confirming that the orientation of NWs impacts cell adhesion. Similarly, the neuronal area (β-tub III area) was increased in R-NWs substrates, which is indicative of higher neuronal branching [57]. This effect might be related to adhesion processes [57]. Studies showed that when using InAs NWs or Si NWs as a substrate, it was possible to obtain higher neurite outgrowth or longer neurites than in a planar control [26,58]. However, we did not observe differences in β-tub III cells between the V-NWs and the control substrates due to the high hydrophobic surfaces present on the V-NWs compared with the R-NWs. Therefore, using NWs substrates can increase the β-tub III area, and this parameter was associated with the orientation. We obtained a higher β-tub III area using R-NWs than V-NWs compared with the control. In accordance with the higher cell adhesion promoted by R-NWs is the increased number of glial cells (GFAP^+^). Intriguingly, the morphology of GFAP^+^ cells was also affected by NWs. In the V-NWs platform, besides reducing glial cell adhesion compared with the R-NWs, an interesting feature when developing electrodes for brain implants [59] augmented the number of stellate shape astrocytes. Three-dimensional scaffolds can favour a stellate shape on glial cells compared with two-dimensional substrates due to simulation in an in vivo model [8,60]. Previous investigations showed that cell volume and processes morphology can change astrocyte function [61]. Furthermore, astrocyte morphology is key to their function and communication with neurons [62]. Astrocyte two-dimensional morphology in vitro becomes stellate when β-adrenergic receptors are activated [63]. It is plausible that these morphological changes are associated with the changes in network signaling we observed in the V-NWs platforms.

In this study, the orientation of NWs in the platform induced a different response in the neuronal activity in terms of frequency. Neuronal networks interfaced with the V-NWs displayed high spontaneous activity when recorded in live imaging compared with the control and the R-NWs. In the R-NWs substrates, we obtained a lower frequency of neuronal activity in the standard saline solution, which increased later due to bicuculline treatment, namely by the removal of synaptic inhibition, reaching the control frequency values. To understand the increase in activity frequency using the V-NWs and the decrease in the R-NWs platform, we calculated the mean correlation coefficient related to neuronal synchronization. This analysis showed no statistical differences in the traces of synchronization between the control, R-NWs, and V-NWs (Appendix A). These changes in neuronal activity might be related to different network compositions due to growth substrates, particularly the number of GABAergic neurons. These cells are in charge of the inhibition of the neuronal network, and we found out that the V-NWs possessed a lower number of GABAergic neurons compared with the control, in accordance with the fact that the frequency of the V-NWs substrates was not affected by bicuculline. There were no significant differences between the control and R-NWs substrates in the number of GABAergic neurons, but in the R-NWs, we observed an increase in the frequency when we applied bicuculline, probably due to a different level of maturation of synaptic inhibition of the R-NWs substrates. In this regard, a previous study showed that InAs NWs substrates orientation influences cell maturation [58]. 

## 5. Conclusions

In summary, we found that the orientation of NWs in the platforms is important and can modulate the density and activity of the interfaced neuronal network. R-NWs produced a higher area of β-tub III. On the contrary, V-NWs platforms produced the same amount of β-tub III as the control, suggesting that the NWs orientation affects cell adhesion. These observations deserve future studies with a deeper study on the impact of NWs orientation on cell morphology. In terms of neuronal activity, R-NWs had a decrease in neuronal activity frequency. However, V-NWs substrates produced an increase in neuronal activity, probably due to a change in the GFAP cell phenotype, confirming the ability of NWs orientation to impact neuronal network properties. NWs orientation supports different GABAergic neuronal phenotypes, with fewer GABAergic cells contributing to network excitability. Future efforts should focus on interface interactions between NWs and the cell membrane using high-quality live imaging and transmission electron microscopy.

## Figures and Tables

**Figure 1 biomolecules-13-00783-f001:**
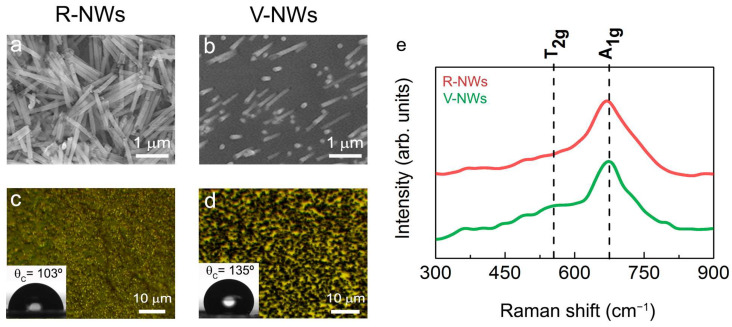
SEM images of electrodeposited iron (**a**) random NWs (R-NWs) and (**b**) vertical NWs (V-NWs). Optical images and contact angles of (**c**) R-NWs and (**d**) V-NWs orientations. (**e**) Raman spectra from 300 cm^−1^ to 900 cm^−1^ for R-NWs and V-NWs substrates. The positions for the active vibrational modes of magnetite are marked with vertical black lines. In-plane Raman intensity images measuring different single Raman spectra were taken at 100 nm with a laser excitation power of 0.25 mW for R-NWs (red) and V-NWs substrates (green).

**Figure 2 biomolecules-13-00783-f002:**
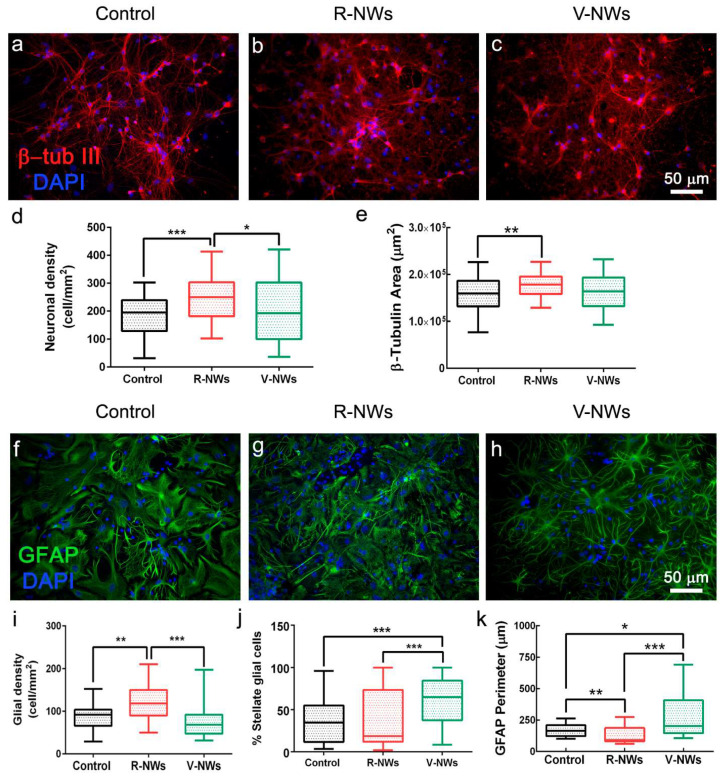
Hippocampal cell densities by NWs orientation. Immunofluorescence micrographs visualizing neurons (**a**–**c**), red anti-β-tub III and glial cells (**f**–**h**), and green anti-GFAP in three different conditions. Nuclei are visualized by DAPI (in blue). Scale bars: 50 μm. Box plot (**d**) summarizes neuronal densities of the control (n = 59 visual fields, 8 samples, n = 3 series of cultures), R-NWs (n = 62 visual fields, 7 samples, n = 3 series of cultures), and V-NWs substrates (n = 59 visual fields, 8 samples, n = 3 series of cultures) and (**e**) β-tub III areas of the control (n = 65 visual fields, 8 samples), R-NWs (n = 69, 7 samples), and V-NWs substrates (n = 68, 8 samples). Box plot (**i**) summarizes the glial densities in the control (n = 45, 6 samples), R-NWs (n = 46, 5 samples), and V-NWs substrates (n = 46, 6 samples). (**j**) The percentage of stellate glial cells in the control (n = 43, 6 samples), R-NWs (n = 44, 5 samples), and V-NWs substrates (n = 43, 6 samples). (**k**) The GFAP perimeter for the substrates tested in the control (n = 51, 6 samples), R-NWs (n = 50, 5 samples), and V-NWs substrates (n = 44, 6 samples). *p* values: * *p* < 0.05, ** *p* < 0.01, and *** *p* < 0.001.

**Figure 3 biomolecules-13-00783-f003:**
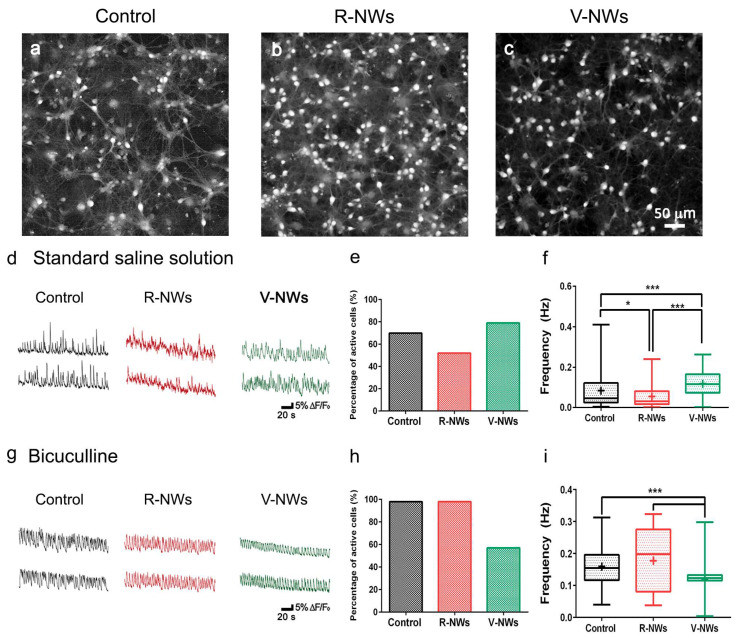
Hippocampal network activity by NWs orientation. Snapshots of representative fields of neuronal cultures grown on the control (**a**), R-NWs (**b**), and V-NWs (**c**) substrates cells are visualized by Oregon-Green 488 BAPTA-1 AM. Representative fluorescence tracings of repetitive Ca^2+^-transients in standard saline solution (**d**) or bicuculline-induced (**g**) recorded in hippocampal cultures of 8–10 DIV grown on the control (black), R-NWs (red), and V-NWs (green) (two sample neurons were selected from the same field). (**e**) The bar plot summarize the percentage of spontaneously active cells in a standard saline solution (control: n = 8 visual fields; R-NWs: n = 6; and V-NWs: n = 6; n = 3 series of cultures). (**f**) Box plot frequencies values for standard saline (control: n = 273 cells, 8 samples; R-NWs: n = 95 cells, 5 samples; and V-NW: n = 506 cells, 5 samples). (**h**) Percentage of spontaneously active cells in bicuculline (control: n = 7; R-NWs: n = 4; and V-NWs: n = 5). (**i**) Frequencies values in bicuculline (control: n = 430 cells, 7 samples; R-NWs: n = 119 cells, 5 samples; and V-NWs: n = 497 cells, 4 samples). *p* values: * *p* < 0.05, and *** *p* < 0.001.

**Figure 4 biomolecules-13-00783-f004:**
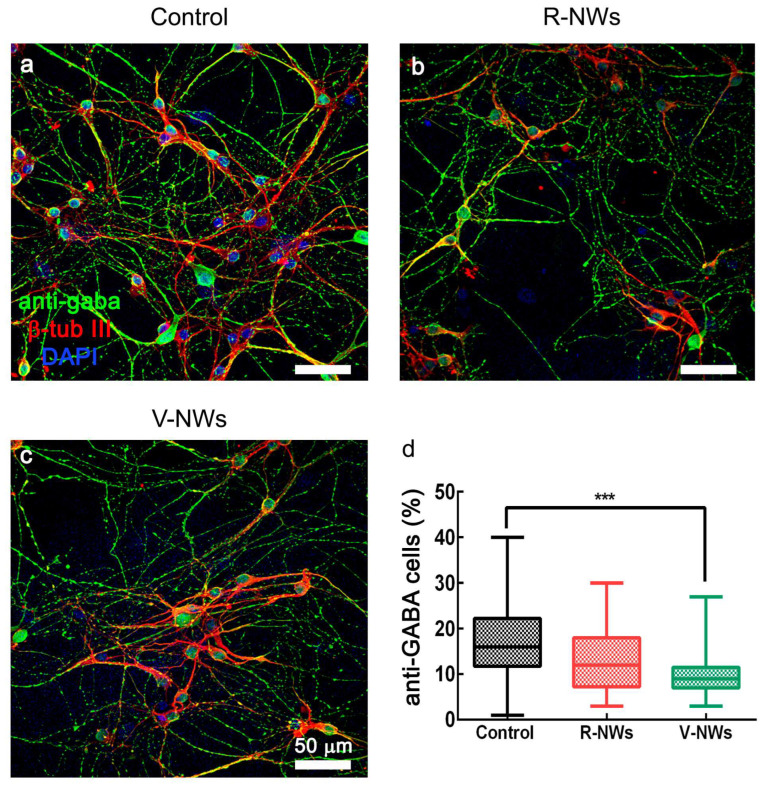
GABAergic neuronal phenotype by NWs orientation. Confocal micrographs show hippocampal cultures grown (8–10 DIV) on (**a**) control, (**b**) R-NWs, and (**c**) V-NWs magnetite platforms immune-stained for GABA (in green), β-tub III (in red), and DAPI (in blue). Scale bar: 50 μm. (**d**) The plot summarizes the percentage of GABA-ergic positive neurons in the control (n = 30 visual fields, 3 samples), R-NWs (n = 28, 3 samples), and V-NWs (n = 29, 3 samples). *p* value: *** *p* < 0.001.

## Data Availability

All the data needed to evaluate the conclusions in the paper are present in the paper and/or the Appendix A. Correspondence and requests for materials should be addressed to B.C.-L. (belencortesllanos@gmail.com), R.R. (rossana.rauti@uniurb.it), and L.B. (laura.ballerini@sissa.it).

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
