# Peer review of "Impact of Magnetite Nanowires on In Vitro Hippocampal Neural Networks"

_biomolecules, 2023, doi:10.3390/biom13050783_

Round 1

Reviewer 1 Report

In this manuscript, two orientation configurations of iron oxide nanowires (random R-NWs and vertical V-RWs) were studied in vitro on a functional brain network to evaluate the modulatory effect of orientation on the activity of emerging synaptic networks. A variety of analyses were performed. The manuscript contains interesting results on the impact on neuronal growth and functionality following orientation modification of NWs.

The introduction could be improved with research after 2019.

Reviewer 2 Report

Manuscript ID: biomolecules-2317845

Title: Impact of magnetite nanowires orientation on morphology and activity of in vitro hippocampal neural networks

Reviewer’s comments

 In this study, the authors cultured hippocampal neurons on magnetite nanowire deposited substrates and investigated the influence of the nanowire orientation on the cellular behavior. My major comments are as the follows:

1.     The purpose and objectives of this study are not clear, what is the problem that the authors intended to solve by using the method and why this nanowire method was chosen are not explained in the manuscript.

2.     The toxicity of the nanowires should be examined, although the biocompatibility of Fe3O4 was regarded as not an issue based on literature; the toxicity of Fe3O4 is tightly related with the density and coating, the authors should test it by themselves in this experiment.

3.     There is no investigation on the density and orientation of the nanowires during the culture, it is quite considerable that these parameters about the nanowire morphology altered due to the cellular activity.

4.     The influence of the nanowire orientation was described as it “regulated” cellular activities as in Fig 3 and 4. I do not think the results showed that the authors could “regulate” the cellular behavior through the nanowire orientation.

5.     The sample number “n” for each experiment should be indicated in the captions of the figures.

6.     The authors explained that the orientation impacted the cell adhesion. It is not clear whether the orientation directly impact cell adhesion or indirectly, for instance, due to the change of the extent of surface hydrophobicity.

7.     The authors claimed that “ Substrates with higher thickness can favour this … “. I cannot understand how this higher thickness of a culture substrate could affect anything on cultured cells.

Reviewer 3 Report

The authors obtained interesting nanostructures. However, the title of the article does not strictly correspond to the results obtained. The conclusions require additional experimental confirmation. Effects on neuronal calcium signaling have not been adequately studied.

1) The ratio of neurons and astrocytes in culture should be clearly shown. For example, it should be indicated in the materials and methods that our cultivation conditions lead to the fact that by the beginning of the experiments, the percentage of neurons in culture was X%, and astrocytes XX%.

2) "In our recordings, spontaneous Ca2+ activity was detected in 70% of cells" Is it 70% of neurons? At similar stages of cultivation, astrocytes are also able to generate spontaneous, although not synchronous, Ca2+ oscillations.

3) Bicuculline should have a clear effect on Ca2+ oscillations, leading to an increase in their amplitude or period of oscillations. Figure 3g does not show the point where bicuculline was added. It is not obvious, based on the two cells presented, that there are changes in the pattern of bicuculline-induced Ca2+ oscillations. There are suspicions that it could be 2 adjacent astrocytes??? It would be acceptable to add 35mM KCl at the end of the experiment, to which only neurons will respond, since they express voltage-gated Ca2+ channels at a high level. A Y scale is needed so that the reader can see the dimension of the oscillation amplitude. Also, the figure should indicate a calcium-sensitive probe for the convenience of the reader of the article. https://pubmed.ncbi.nlm.nih.gov/32727246/

4) Can a decrease in the number of GABA+ neurons indicate a decrease in the inhibitory component of the network? Is it then possible to speak of a greater tendency of the network that grows on this substrate to epileptiform activity?

5) Did the authors consider the possible cytotoxic effect of their nanostructures? For example, ferroptosis

6) The conclusion is not obvious. The effects of nanomaterials on cell morphology require analysis of neurite outgrowth, analysis of the expression of genes encoding protein-regulators of network differentiation, etc.

Round 2

Reviewer 2 Report

no further comments and suggestions.

Reviewer 3 Report

All my comments have been taken into account. article can be accepted for publication